# Safe Exploration in Finite Markov Decision Processes with Gaussian Processes

**Matteo Turchetta**
ETH Zurich
matteotu@ethz.ch

**Felix Berkenkamp**
ETH Zurich
befelix@ethz.ch

**Andreas Krause**
ETH Zurich
krausea@ethz.ch

## Abstract

In classical reinforcement learning agents accept arbitrary short term loss for long term gain when exploring their environment. This is infeasible for safety critical applications such as robotics, where even a single unsafe action may cause system failure or harm the environment. In this paper, we address the problem of safely exploring finite Markov decision processes (MDP). We define safety in terms of an a priori unknown safety constraint that depends on states and actions and satisfies certain regularity conditions expressed via a Gaussian process prior. We develop a novel algorithm, SAFEMDP, for this task and prove that it completely explores the safely reachable part of the MDP without violating the safety constraint. To achieve this, it cautiously explores safe states and actions in order to gain statistical confidence about the safety of unvisited state-action pairs from noisy observations collected while navigating the environment. Moreover, the algorithm explicitly considers reachability when exploring the MDP, ensuring that it does not get stuck in any state with no safe way out. We demonstrate our method on digital terrain models for the task of exploring an unknown map with a rover.

## 1 Introduction

Today's robots are required to operate in variable and often unknown environments. The traditional solution is to specify all potential scenarios that a robot may encounter during operation *a priori*. This is time consuming or even infeasible. As a consequence, robots need to be able to learn and adapt to unknown environments autonomously [10, 2]. While exploration algorithms are known, safety is still an open problem in the development of such systems [18]. In fact, most learning algorithms allow robots to make unsafe decisions during exploration. This can damage the platform or its environment.

In this paper, we provide a solution to this problem and develop an algorithm that enables agents to safely and autonomously explore unknown environments. Specifically, we consider the problem of exploring a Markov decision process (MDP), where it is *a priori* unknown which state-action pairs are safe. Our algorithm cautiously explores this environment without taking actions that are unsafe or may render the exploring agent stuck.

**Related Work.** Safe exploration is an open problem in the reinforcement learning community and several definitions of safety have been proposed [16]. In risk-sensitive reinforcement learning, the goal is to maximize the expected return for the worst case scenario [5]. However, these approaches only minimize risk and do not treat safety as a hard constraint. For example, Geibel and Wysotzki [7] define risk as the probability of driving the system to a previously known set of undesirable states. The main difference to our approach is that we do not assume the undesirable states to be known *a priori*. Garcia and Fernández [6] propose to ensure safety by means of a backup policy; that is, a policy that is known to be safe in advance. Our approach is different, since it does not require a backup policy but only a set of initially safe states from which the agent starts to explore. Another approach that makes use of a backup policy is shown by Hans et al. [9], where safety is defined in terms of a minimum reward, which is learned from data.

Moldovan and Abbeel [14] provide probabilistic safety guarantees at every time step by optimizing over ergodic policies; that is, policies that let the agent recover from any visited state. This approach needs to solve a large linear program at every time step, which is computationally demanding even for small state spaces. Nevertheless, the idea of ergodicity also plays an important role in our method. In the control community, safety is mostly considered in terms of stability or constraint satisfaction of controlled systems. Akametalu et al. [1] use reachability analysis to ensure stability under the assumption of bounded disturbances. The work in [3] uses robust control techniques in order to ensure robust stability for model uncertainties, while the uncertain model is improved.

Another field that has recently considered safety is Bayesian optimization [13]. There, in order to find the global optimum of an *a priori* unknown function [21], regularity assumptions in form of a Gaussian process (GP) [17] prior are made. The corresponding GP posterior distribution over the unknown function is used to guide evaluations to informative locations. In this setting, safety centered approaches include the work of Sui et al. [22] and Schreiter et al. [20], where the goal is to find the safely reachable optimum without violating an *a priori* unknown safety constraint at any evaluation. To achieve this, the function is cautiously explored, starting from a set of points that is known to be safe initially. The method in [22] was applied to the field of robotics to safely optimize the controller parameters of a quadrotor vehicle [4]. However, they considered a bandit setting, where at each iteration any arm can be played. In contrast, we consider exploring an MDP, which introduces restrictions in terms of reachability that have not been considered in Bayesian optimization before.

**Contribution.** We introduce SAFEMDP, a novel algorithm for safe exploration in MDPs. We model safety via an *a priori* unknown constraint that depends on state-action pairs. Starting from an initial set of states and actions that are known to satisfy the safety constraint, the algorithm exploits the regularity assumptions on the constraint function in order to determine if nearby, unvisited states are safe. This leads to safe exploration, where only state-actions pairs that are known to fulfil the safety constraint are evaluated. The main contribution consists of extending the work on safe Bayesian optimization in [22] from the bandit setting to deterministic, finite MDPs. In order to achieve this, we explicitly consider not only the safety constraint, but also the reachability properties induced by the MDP dynamics. We provide a full theoretical analysis of the algorithm. It provably enjoys similar safety guarantees in terms of ergodicity as discussed in [14], but at a reduced computational cost. The reason for this is that our method separates safety from the reachability properties of the MDP. Beyond this, we prove that SAFEMDP is able to fully explore the safely reachable region of the MDP, without getting stuck or violating the safety constraint with high probability. To the best of our knokwledge, this is the first full exploration result in MDPs subject to a safety constraint. We validate our method on an exploration task, where a rover has to explore an *a priori* unknown map.

## 2 Problem Statement

In this section, we define our problem and assumptions. The unknown environment is modeled as a finite, deterministic MDP [23]. Such a MDP is a tuple $\langle \mathcal{S}, \mathcal{A}(\cdot), f(\mathbf{s}, a), r(\mathbf{s}, a) \rangle$ with a finite set of states $\mathcal{S}$, a set of state-dependent actions $\mathcal{A}(\cdot)$, a known, deterministic transition model $f(\mathbf{s}, a)$, and reward function $r(\mathbf{s}, a)$. In the typical reinforcement learning framework, the goal is to maximize the cumulative reward. In this paper, we consider the problem of safely exploring the MDP. Thus, instead of aiming to maximize the cumulative rewards, we define $r(\mathbf{s}, a)$ as an *a priori* unknown safety feature. Although $r(\mathbf{s}, a)$ is unknown, we make regularity assumptions about it to make the problem tractable. When traversing the MDP, at each discrete time step, $k$, the agent has to decide which action and thereby state to visit next. We assume that the underlying system is safety-critical and that for any visited state-action pair, $(\mathbf{s}_k, a_k)$, the unknown, associated safety feature, $r(\mathbf{s}_k, a_k)$, must be above a safety threshold, $h$. While the assumption of deterministic dynamics does not hold for general MDPs, in our framework, uncertainty about the environment is captured by the safety feature. If requested, the agent can obtain noisy measurements of the safety feature, $r(\mathbf{s}_k, a_k)$, by taking action $a_k$ in state $\mathbf{s}_k$. The index $t$ is used to index measurements, while $k$ denotes movement steps. Typically $k \gg t$.

It is hopeless to achieve the goal of safe exploration unless the agent starts in a safe location. Hence, we assume that the agent stays in an initial set of state action pairs, $S_0$, that is known to be safe *a priori*. The goal is to identify the maximum safely reachable region starting from $S_0$, without visiting any unsafe states. For clarity of exposition, we assume that safety depends on states only; that is, $r(\mathbf{s}, a) = r(\mathbf{s})$. We provide an extension to safety features that also depend on actions in Sec. 3.

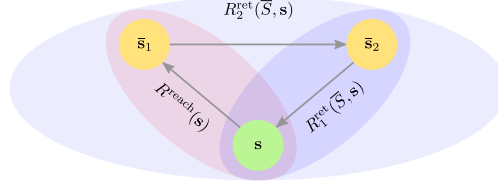

Figure 1: Illustration of the set operators with $\overline{S} = \{\bar{\mathbf{s}}_1, \bar{\mathbf{s}}_2\}$. The set $S = \{\mathbf{s}\}$ can be reached from $\bar{\mathbf{s}}_2$ in one step and from $\bar{\mathbf{s}}_1$ in two steps, while only the state $\bar{\mathbf{s}}_1$ can be reached from $\mathbf{s}$. Visiting $\bar{\mathbf{s}}_1$ is safe; that is, it is above the safety threshold, is reachable, and there exists a safe return path through $\mathbf{s}_2$.

**Assumptions on the reward function** Ensuring that all visited states are safe without any prior knowledge about the safety feature is an impossible task (e.g., if the safety feature is discontinuous). However, many practical safety features exhibit some regularity, where similar states will lead to similar values of $r$.

In the following, we assume that $\mathcal{S}$ is endowed with a positive definite kernel function $k(\cdot, \cdot)$ and that the function $r(\cdot)$ has bounded norm in the associated Reproducing Kernel Hilbert Space (RKHS) [19]. The norm induced by the inner product of the RKHS indicates the smoothness of functions with respect to the kernel. This assumption allows us to model $r$ as a GP [21], $r(\mathbf{s}) \sim \mathcal{GP}(\mu(\mathbf{s}), k(\mathbf{s}, \mathbf{s}'))$. A GP is a probability distribution over functions that is fully specified by its mean function $\mu(\mathbf{s})$ and its covariance function $k(\mathbf{s}, \mathbf{s}')$. The randomness expressed by this distribution captures our uncertainty about the environment. We assume $\mu(\mathbf{s}) = 0$ for all $\mathbf{s} \in \mathcal{S}$, without loss of generality. The posterior distribution over $r(\cdot)$ can be computed analytically, based on $t$ measurements at states $D_t = \{\mathbf{s}_1, \ldots, \mathbf{s}_t\} \subseteq \mathcal{S}$ with measurements, $\mathbf{y}_t = [r(\mathbf{s}_1) + \omega_1 \ldots r(\mathbf{s}_t) + \omega_t]^{\mathrm{T}}$, that are corrupted by zero-mean Gaussian noise, $\omega_t \sim \mathcal{N}(0, \sigma^2)$. The posterior is a $\mathcal{GP}$ distribution with mean $\mu_t(\mathbf{s}) = \mathbf{k}_t(\mathbf{s})^{\mathrm{T}}(\mathbf{K}_t + \sigma^2 \mathbf{I})^{-1}\mathbf{y}_t$, variance $\sigma_t(\mathbf{s}) = k_t(\mathbf{s}, \mathbf{s})$, and covariance $k_t(\mathbf{s}, \mathbf{s}') = k(\mathbf{s}, \mathbf{s}') - \mathbf{k}_t(\mathbf{s})^{\mathrm{T}}(\mathbf{K}_t + \sigma^2 \mathbf{I})^{-1}\mathbf{k}_t(\mathbf{s}')$, where $\mathbf{k}_t(\mathbf{s}) = [k(\mathbf{s}_1, \mathbf{s}), \ldots, k(\mathbf{s}_t, \mathbf{s})]^{\mathrm{T}}$ and $\mathbf{K}_t$ is the positive definite kernel matrix, $[k(\mathbf{s}, \mathbf{s}')]_{\mathbf{s}, \mathbf{s}' \in D_t}$. The identity matrix is denoted by $\mathbf{I} \in \mathbb{R}^{t \times t}$.

We also assume $L$-Lipschitz continuity of the safety function with respect to some metric $d(\cdot, \cdot)$ on $\mathcal{S}$. This is guaranteed by many commonly used kernels with high probability [21, 8].

**Goal** In this section, we define the goal of safe exploration. In particular, we ask what the best that any algorithm may hope to achieve is. Since we only observe noisy measurements, it is impossible to know the underlying safety function $r(\cdot)$ exactly after a finite number of measurements. Instead, we consider algorithms that only have knowledge of $r(\cdot)$ up to some statistical confidence $\epsilon$. Based on this confidence within some safe set $S$, states with small distance to $S$ can be classified to satisfy the safety constraint using the Lipschitz continuity of $r(\cdot)$. The resulting set of safe states is

$$R_\epsilon^{\mathrm{safe}}(S) = S \cup \{\mathbf{s} \in \mathcal{S} \mid \exists \mathbf{s}' \in S \colon r(\mathbf{s}') - \epsilon - Ld(\mathbf{s}, \mathbf{s}') \geq h\}, \tag{1}$$

which contains states that can be classified as safe given the information about the states in $S$. While (1) considers the safety constraint, it does not consider any restrictions put in place by the structure of the MDP. In particular, we may not be able to visit every state in $R_\epsilon^{\mathrm{safe}}(S)$ without visiting an unsafe state first. As a result, the agent is further restricted to

$$R^{\mathrm{reach}}(S) = S \cup \{\mathbf{s} \in \mathcal{S} \mid \exists \mathbf{s}' \in S, a \in \mathcal{A}(\mathbf{s}') \colon \mathbf{s} = f(\mathbf{s}', a)\}, \tag{2}$$

the set of all states that can be reached starting from the safe set in one step. These states are called the one-step safely reachable states. However, even restricted to this set, the agent may still get stuck in a state without any safe actions. We define

$$R^{\mathrm{ret}}(S, \overline{S}) = \overline{S} \cup \{\mathbf{s} \in S \mid \exists a \in \mathcal{A}(\mathbf{s}) \colon f(\mathbf{s}, a) \in \overline{S}\} \tag{3}$$

as the set of states that are able to return to a set $\overline{S}$ through some other set of states, $S$, in one step. In particular, we care about the ability to return to a certain set through a set of safe states $S$. Therefore, these are called the one-step safely returnable states. In general, the return routes may require taking more than one action, see Fig. 1. The $n$-step returnability operator $R_n^{\mathrm{ret}}(S, \overline{S}) = R^{\mathrm{ret}}(S, R_{n-1}^{\mathrm{ret}}(S, \overline{S}))$ with $R_1^{\mathrm{ret}}(S, \overline{S}) = R^{\mathrm{ret}}(S, \overline{S})$ considers these longer return routes by repeatedly applying the return operator, $R^{\mathrm{ret}}$ in (3), $n$ times. The limit $\overline{R}^{\mathrm{ret}}(S, \overline{S}) = \lim_{n \to \infty} R_n^{\mathrm{ret}}(S, \overline{S})$ contains all the states that can reach the set $\overline{S}$ through an arbitrarily long path in $S$.

---
**Algorithm 1** Safe exploration in MDPs (**SafeMDP**)
---
**Inputs:** states $\mathcal{S}$, actions $\mathcal{A}$, transition function $f(\mathbf{s}, a)$, kernel $k(\mathbf{s}, \mathbf{s}')$, Safety
       threshold $h$, Lipschitz constant $L$, Safe seed $S_0$.
$C_0(s) \leftarrow [h, \infty)$ for all $s \in S_0$
**for** $t = 1, 2, \ldots$ **do**
    $S_t \leftarrow \{\mathbf{s} \in \mathcal{S} \mid \exists \mathbf{s}' \in \hat{S}_{t-1} \colon l_t(\mathbf{s}') - Ld(\mathbf{s}, \mathbf{s}') \geq h\}$
    $\hat{S}_t \leftarrow \{\mathbf{s} \in S_t \mid \mathbf{s} \in R^{\mathrm{reach}}(\hat{S}_{t-1}), \mathbf{s} \in \overline{R}^{\mathrm{ret}}(S_t, \hat{S}_{t-1})\}$
    $G_t \leftarrow \{\mathbf{s} \in \hat{S}_t \mid g_t(\mathbf{s}) > 0\}$
    $\mathbf{s}_t \leftarrow \operatorname{argmax}_{\mathbf{s} \in G_t} w_t(\mathbf{s})$
    Safe Dijkstra in $\hat{S}_t$ from $\mathbf{s}_{t-1}$ to $\mathbf{s}_t$
    Update GP with $\mathbf{s}_t$ and $y_t \leftarrow r(\mathbf{s}_t) + \omega_t$
    **if** $G_t = \emptyset$ **or** $\max\limits_{\mathbf{s} \in G_t} w_t(\mathbf{s}) \leq \epsilon$ **then** Break
---

For safe exploration of MDPs, all of the above are requirements; that is, any state that we may want to visit needs to be safe (satisfy the safety constraint), reachable, and we must be able to return to safe states from this new state. Thus, any algorithm that aims to safely explore an MDP is only allowed to visit states in

$$R_\epsilon(S) = R_\epsilon^{\mathrm{safe}}(S) \cap R^{\mathrm{reach}}(S) \cap \overline{R}^{\mathrm{ret}}(R_\epsilon^{\mathrm{safe}}(S), S), \tag{4}$$

which is the intersection of the three safety-relevant sets. Given a safe set $S$ that fulfills the safety requirements, $\overline{R}^{\mathrm{ret}}(R_\epsilon^{\mathrm{safe}}(S), S)$ is the set of states from which we can return to $S$ by only visiting states that can be classified as above the safety threshold. By including it in the definition of $R_\epsilon(S)$, we avoid the agent getting stuck in a state without an action that leads to another safe state to take.

Given knowledge about the safety feature in $S$ up to $\epsilon$ accuracy thus allows us to expand the set of safe ergodic states to $R_\epsilon(S)$. Any algorithm that has the goal of exploring the state space should consequently explore these newly available safe states and gain new knowledge about the safety feature to potentially further enlargen the safe set. The safe set after $n$ such expansions can be found by repeatedly applying the operator in (4): $R_\epsilon^n(S) = R_\epsilon(R_\epsilon^{n-1}(S))$ with $R_\epsilon^1 = R_\epsilon(S)$. Ultimately, the size of the safe set is bounded by surrounding unsafe states or the number of states in $\mathcal{S}$. As a result, the biggest set that any algorithm may classify as safe without visiting unsafe states is given by taking the limit, $\overline{R}_\epsilon(S) = \lim_{n \to \infty} R_\epsilon^n(S)$.

Thus, given a tolerance level $\epsilon$ and an initial safe seed set $S_0$, $\overline{R}_\epsilon(S_0)$ is the set of states that any algorithm may hope to classify as safe. Let $S_t$ denote the set of states that an algorithm determines to be safe at iteration $t$. In the following, we will refer to complete, safe exploration whenever an algorithm fulfills $\overline{R}_\epsilon(S_0) \subseteq \lim_{t \to \infty} S_t \subseteq \overline{R}_0(S_0)$; that is, the algorithm classifies every safely reachable state up to $\epsilon$ accuracy as safe, without misclassification or visiting unsafe states.

## 3 SAFEMDP Algorithm

We start by giving a high level overview of the method. The SAFEMDP algorithm relies on a GP model of $r$ to make predictions about the safety feature and uses the predictive uncertainty to guide the safe exploration. In order to guarantee safety, it maintains two sets. The first set, $S_t$, contains all states that can be classified as satisfying the safety constraint using the GP posterior, while the second one, $\hat{S}_t$, additionally considers the ability to reach points in $S_t$ and the ability to safely return to the previous safe set, $\hat{S}_{t-1}$. The algorithm ensures safety and ergodicity by only visiting states in $\hat{S}_t$. In order to expand the safe region, the algorithm visits states in $G_t \subseteq \hat{S}_t$, a set of candidate states that, if visited, could expand the safe set. Specifically, the algorithm selects the most uncertain state in $G_t$, which is the safe state that we can gain the most information about. We move to this state via the shortest safe path, which is guaranteed to exist (Lemma 2). The algorithm is summarized in Algorithm 1.

**Initialization.** The algorithm relies on an initial safe set $S_0$ as a starting point to explore the MDP. These states must be safe; that is, $r(s) \geq h$, for all $s \in S_0$. They must also fulfill the reachability and returnability requirements from Sec. 2. Consequently, for any two states, $\mathbf{s}, \mathbf{s}' \in S_0$, there must exist a path in $S_0$ that connects them: $\mathbf{s}' \in \overline{R}^{\mathrm{ret}}(S_0, \{\mathbf{s}\})$. While this may seem restrictive, the requirement is, for example, fulfilled by a single state with an action that leads back to the same state.

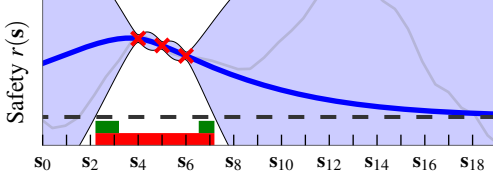

(a) States are classified as safe (above the safety constraint, dashed line) according to the confidence intervals of the GP model (red bar). States in the green bar can expand the safe set if sampled, $G_t$.

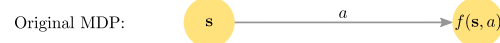

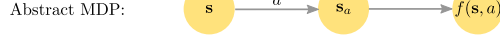

(b) Modified MDP model that is used to encode safety features that depend on actions. In this model, actions lead to abstract action-states $\mathbf{s}_a$, which only have one available action that leads to $f(\mathbf{s}, a)$.

**Classification.** In order to safely explore the MDP, the algorithm must determine which states are safe without visiting them. The regularity assumptions introduced in Sec. 2 allow us to model the safety feature as a $\mathcal{GP}$, so that we can use the uncertainty estimate of the GP model in order to determine a confidence interval within which the true safety function lies with high probability. For every state $\mathbf{s}$, this confidence interval has the form $Q_t(\mathbf{s}) = \left[\mu_{t-1}(\mathbf{s}) \pm \sqrt{\beta_t}\sigma_{t-1}(\mathbf{s})\right]$, where $\beta_t$ is a positive scalar that determines the amplitude of the interval. We discuss how to select $\beta_t$ in Sec. 4.

Rather than defining high probability bounds on the values of $r(\mathbf{s})$ directly in terms of $Q_t$, we consider the intersection of the sets $Q_t$ up to iteration $t$, $C_t(\mathbf{s}) = Q_t(\mathbf{s}) \cap C_{t-1}(\mathbf{s})$ with $C_0(\mathbf{s}) = [h, \infty]$ for safe states $\mathbf{s} \in S_0$ and $C_0(\mathbf{s}) = \mathbb{R}$ otherwise. This choice ensures that set of states that we classify as safe does not shrink over iterations and is justified by the selection of $\beta_t$ in Sec. 4. Based on these confidence intervals, we define a lower bound, $l_t(\mathbf{s}) = \min C_t(\mathbf{s})$, and upper bound, $u_t(\mathbf{s}) = \max C_t(\mathbf{s})$, on the values that the safety features $r(\mathbf{s})$ are likely to take based on the data obtained up to iteration $t$. Based on these lower bounds, we define

$$S_t = \left\{ \mathbf{s} \in \mathcal{S} \,|\, \exists \mathbf{s}' \in \hat{S}_{t-1} : \, l_t(\mathbf{s}') - Ld(\mathbf{s}, \mathbf{s}') \geq h \right\} \tag{5}$$

as the set of states that fulfill the safety constraint on $r$ with high probability by using the Lipschitz constant to generalize beyond the current safe set. Based on this classification, the set of ergodic safe states is the set of states that achieve the safety threshold and, additionally, fulfill the reachability and returnability properties discussed in Sec. 2:

$$\hat{S}_t = \left\{ \mathbf{s} \in S_t \,|\, \mathbf{s} \in R^{\text{reach}}(\hat{S}_{t-1}) \cap \overline{R}^{\text{ret}}(S_t, \hat{S}_{t-1}) \right\}. \tag{6}$$

**Expanders.** With the set of safe states defined, the task of the algorithm is to identify and explore states that might expand the set of states that can be classified as safe. We use the uncertainty estimate in the GP in order to define an optimistic set of expanders,

$$G_t = \{\mathbf{s} \in \hat{S}_t \,|\, g_t(\mathbf{s}) > 0\}, \tag{7}$$

where $g_t(\mathbf{s}) = \left|\{\mathbf{s}' \in \mathcal{S} \setminus S_t \,|\, u_t(\mathbf{s}) - Ld(\mathbf{s}, \mathbf{s}') \geq h\}\right|$. The function $g_t(\mathbf{s})$ is positive whenever an optimistic measurement at $\mathbf{s}$, equal to the upper confidence bound, $u_t(\mathbf{s})$, would allow us to determine that a previously unsafe state indeed has value $r(\mathbf{s}')$ above the safety threshold. Intuitively, sampling $\mathbf{s}$ might lead to the expansion of $S_t$ and thereby $\hat{S}_t$. The set $G_t$ explicitly considers the expansion of the safe set as exploration goal, see Fig. 2a for a graphical illustration of the set.

**Sampling and shortest safe path.** The remaining part of the algorithm is concerned with selecting safe states to evaluate and finding a safe path in the MDP that leads towards them. The goal is to visit states that allow the safe set to expand as quickly as possible, so that we do not waste resources when exploring the MDP. We use the GP posterior uncertainty about the states in $G_t$ in order to make this choice. At each iteration $t$, we select as next target sample the state with the highest variance in $G_t$, $\mathbf{s}_t = \text{argmax}_{\mathbf{s} \in G_t} \, w_t(\mathbf{s})$, where $w_t(\mathbf{s}) = u_t(\mathbf{s}) - l_t(\mathbf{s})$. This choice is justified, because while all points in $G_t$ are safe and can potentially enlarge the safe set, based on one noisy sample we can gain the most information from the state that we are the most uncertain about. This design choice maximizes the knowledge acquired with every sample but can lead to long paths between measurements within the safe region. Given $\mathbf{s}_t$, we use Dijkstra's algorithm within the set $\hat{S}_t$ in order to find the shortest safe path to the target from the current state, $s_{t-1}$. Since we require reachability and returnability for all safe states, such a path is guaranteed to exist. We terminate the algorithm when we reach the desired accuracy; that is, $\text{argmax}_{\mathbf{s} \in G_t} \, w_t(\mathbf{s}) \leq \epsilon$.

**Action-dependent safety.** So far, we have considered safety features that only depend on the states, $r(\mathbf{s})$. In general, safety can also depend on the actions, $r(\mathbf{s}, a)$. In this section, we introduce a

modified MDP that captures these dependencies without modifying the algorithm. The modified MDP is equivalent to the original one in terms of dynamics, $f(\mathbf{s}, a)$. However, we introduce additional action-states $\mathbf{s}_a$ for each action in the original MDP. When we start in a state $\mathbf{s}$ and take action $a$, we first transition to the corresponding action-state and from there transition to $f(\mathbf{s}, a)$ deterministically. This model is illustrated in Fig. 2b. Safety features that depend on action-states, $s_a$, are equivalent to action-dependent safety features. The SAFEMDP algorithm can be used on this modified MDP without modification. See the experiments in Sec. 5 for an example.

## 4 Theoretical Results

The safety and exploration aspects of the algorithm that we presented in the previous section rely on the correctness of the confidence intervals $C_t(\mathbf{s})$. In particular, they require that the true value of the safety feature, $r(\mathbf{s})$, lies within $C_t(\mathbf{s})$ with high probability for all $\mathbf{s} \in \mathcal{S}$ and all iterations $t > 0$. Furthermore, these confidence intervals have to shrink sufficiently fast over time. The probability of $r$ taking values within the confidence intervals depends on the scaling factor $\beta_t$. This scaling factor trades off conservativeness in the exploration for the probability of unsafe states being visited. Appropriate selection of $\beta_t$ has been studied by Srinivas et al. [21] in the multi-armed bandit setting. Even though our framework is different, their setting can be applied to our case. We choose,

$$\beta_t = 2B + 300\gamma_t \log^3(t/\delta), \tag{8}$$

where $B$ is the bound on the RKHS norm of the function $r(\cdot)$, $\delta$ is the probability of visiting unsafe states, and $\gamma_t$ is the maximum mutual information that can be gained about $r(\cdot)$ from $t$ noisy observations; that is, $\gamma_t = \max_{|A| \leq t} I(r, \mathbf{y}_A)$. The information capacity $\gamma_t$ has a sublinear dependence on $t$ for many commonly used kernels [21]. The choice of $\beta_t$ in (8) is justified by the following Lemma, which follows from [21, Theorem 6]:

**Lemma 1.** *Assume that $\|r\|_k^2 \leq B$, and that the noise $\omega_t$ is zero-mean conditioned on the history, as well as uniformly bounded by $\sigma$ for all $t > 0$. If $\beta_t$ is chosen as in (8), then, for all $t > 0$ and all $\mathbf{s} \in \mathcal{S}$, it holds with probability at least $1 - \delta$ that $r(\mathbf{s}) \in C_t(\mathbf{s})$.*

This Lemma states that, for $\beta_t$ as in (8), the safety function $r(\mathbf{s})$ takes values within the confidence intervals $C(\mathbf{s})$ with high probability. Now we show that the the safe shortest path problem has always a solution:

**Lemma 2.** *Assume that $S_0 \neq \emptyset$ and that for all states, $\mathbf{s}, \mathbf{s}' \in S_0$, $\mathbf{s} \in \overline{R}^{\mathrm{ret}}(S_0, \{\mathbf{s}'\})$. Then, when using Algorithm 1 under the assumptions in Theorem 1, for all $t > 0$ and for all states, $\mathbf{s}, \mathbf{s}' \in \hat{S}_t$, $\mathbf{s} \in \overline{R}^{\mathrm{ret}}(S_t, \{\mathbf{s}'\})$.*

This lemma states that, given an initial safe set that fulfills the initialization requirements, we can always find a policy that drives us from any state in $\hat{S}_t$ to any other state in $\hat{S}_t$ without leaving the set of safe states, $S_t$. Lemmas 1 and 2 have a key role in ensuring safety during exploration and, thus, in our main theoretical result:

**Theorem 1.** *Assume that $r(\cdot)$ is L-Lipschitz continuous and that the assumptions of Lemma 1 hold. Also, assume that $S_0 \neq \emptyset$, $r(\mathbf{s}) \geq h$ for all $\mathbf{s} \in S_0$, and that for any two states, $\mathbf{s}, \mathbf{s}' \in S_0$, $\mathbf{s}' \in \overline{R}^{\mathrm{ret}}(S_0, \{\mathbf{s}\})$. Choose $\beta_t$ as in (8). Then, with probability at least $1 - \delta$, we have $r(\mathbf{s}) \geq h$ for any $\mathbf{s}$ along any state trajectory induced by Algorithm 1 on an MDP with transition function $f(\mathbf{s}, a)$. Moreover, let $t^*$ be the smallest integer such that $\frac{t^*}{\beta_{t^*} \gamma_{t^*}} \geq \frac{C |\overline{R}_0(S_0)|}{\epsilon^2}$, with $C = 8/\log(1 + \sigma^{-2})$. Then there exists a $t_0 \leq t^*$ such that, with probability at least $1 - \delta$, $\overline{R}_\epsilon(S_0) \subseteq \hat{S}_{t_0} \subseteq \overline{R}_0(S_0)$.*

Theorem 1 states that Algorithm 1 performs safe and complete exploration of the state space; that is, it explores the maximum reachable safe set without visiting unsafe states. Moreover, for any desired accuracy $\epsilon$ and probability of failure $\delta$, the safely reachable region can be found within a finite number of observations. This bound depends on the information capacity $\gamma_t$, which in turn depends on the kernel. If the safety feature is allowed to change rapidly across states, the information capacity will be larger than if the safety feature was smooth. Intuitively, the less prior knowledge the kernel encodes, the more careful we have to be when exploring the MDP, which requires more measurements.

# 5 Experiments

In this section, we demonstrate Algorithm 1 on an exploration task. We consider the setting in [14], the exploration of the surface of Mars with a rover. The code for the experiments is available at http://github.com/befelix/SafeMDP.

For space exploration, communication delays between the rover and the operator on Earth can be prohibitive. Thus, it is important that the robot can act autonomously and explore the environment without risking unsafe behavior. For the experiment, we consider the *Mars Science Laboratory* (MSL) [11], a rover deployed on Mars. Due to communication delays, the MSL can travel 20 meters before it can obtain new instructions from an operator. It can climb a maximum slope of $30°$ [15, Sec. 2.1.3]. In our experiments we use digital terrain models of the surface of Mars from the *High Resolution Imaging Science Experiment* (HiRISE), which have a resolution of one meter [12].

As opposed to the experiments considered in [14], we do not have to subsample or smoothen the data in order to achieve good exploration results. This is due to the flexibility of the GP framework that considers noisy measurements. Therefore, every state in the MDP represents a $d \times d$ square area with $d = 1\,\mathrm{m}$, as opposed to $d = 20\,\mathrm{m}$ in [14].

At every state, the agent can take one of four actions: *up*, *down*, *left*, and *right*. If the rover attempts to climb a slope that is steeper than $30°$, it fails and may be damaged. Otherwise it moves deterministically to the desired neighboring state. In this setting, we define safety over state transitions by using the extension introduced in Sec. 3. The safety feature over the transition from $\mathbf{s}$ to $\mathbf{s}'$ is defined in terms of height difference between the two states, $H(\mathbf{s}) - H(\mathbf{s}')$. Given the maximum slope of $\alpha = 30°$ that the rover can climb, the safety threshold is set at a conservative $h = -d\tan(25°)$. This encodes that it is unsafe for the robot to climb hills that are too steep. In particular, while the MDP dynamics assume that Mars is flat and every state can be reached, the safety constraint depends on the *a priori* unknown heights. Therefore, under the prior belief, it is unknown which transitions are safe.

We model the height distribution, $H(\mathbf{s})$, as a GP with a Matérn kernel with $\nu = 5/2$. Due to the limitation on the grid resolution, tuning of the hyperparameters is necessary to achieve both safety and satisfactory exploration results. With a finer resolution, more cautious hyperparameters would also be able to generalize to neighbouring states. The lengthscales are set to $14.5\,\mathrm{m}$ and the prior standard deviation of heights is $10\,\mathrm{m}$. We assume a noise standard deviation of $0.075\,\mathrm{m}$. Since the safety feature of each state transition is a linear combination of heights, the GP model of the heights induces a GP model over the differences of heights, which we use to classify whether state transitions fulfill the safety constraint. In particular, the safety depends on the direction of travel, that is, going downhill is possible, while going uphill might be unsafe.

Following the recommendations in [22], in our experiments we use the GP confidence intervals $Q_t(\mathbf{s})$ directly to determine the safe set $S_t$. As a result, the Lipschitz constant is only used to determine expanders in $G$. Guaranteeing safe exploration with high probability over multiple steps leads to conservative behavior, as every step beyond the set that is known to be safe decreases the 'probability budget' for failure. In order to demonstrate that safety can be achieved empirically using less conservative parameters than those suggested by Theorem 1, we fix $\beta_t$ to a constant value, $\beta_t = 2, \forall t \geq 0$. This choice aims to guarantee safety per iteration rather than jointly over all the iterations. The same assumption is used in [14].

We compare our algorithm to several baselines. The first one considers both the safety threshold and the ergodicity requirements but neglects the expanders. In this setting, the agent samples the most uncertain safe state transaction, which corresponds to the safe Bayesian optimization framework in [20]. We expect the exploration to be safe, but less efficient than our approach. The second baseline considers the safety threshold, but does not consider ergodicity requirements. In this setting, we expect the rover's behavior to fulfill the safety constraint and to never attempt to climb steep slopes, but it may get stuck in states without safe actions. The third method uses the unconstrained Bayesian optimization framework in order to explore new states, without safety requirements. In this setting, the agent tries to obtain measurements from the most uncertain state transition over the entire space, rather than restricting itself to the safe set. In this case, the rover can easily get stuck and may also incur failures by attempting to climb steep slopes. Last, we consider a random exploration strategy, which is similar to the $\epsilon$-greedy exploration strategies that are widely used in reinforcement learning.

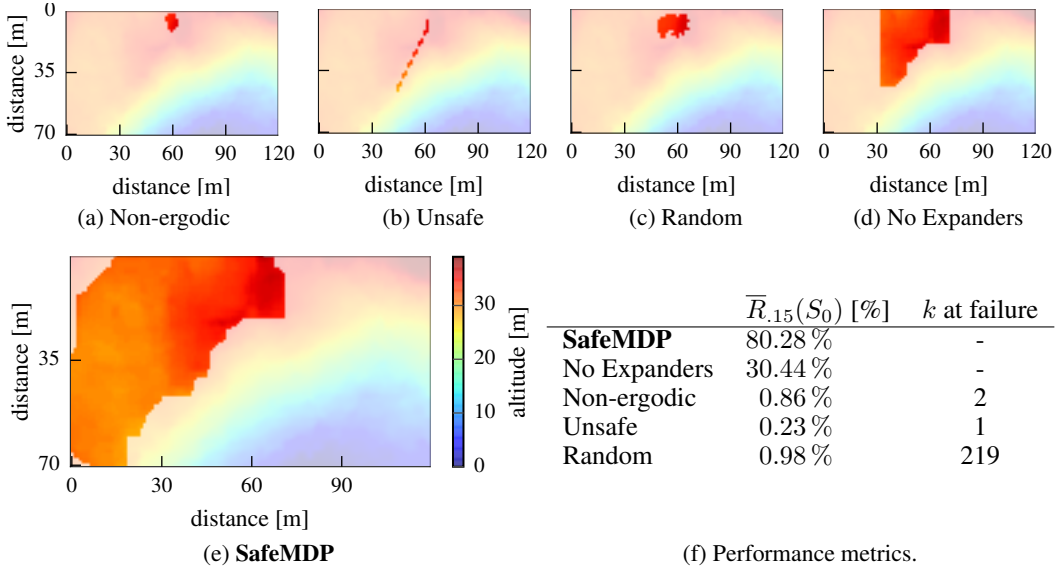

Figure 2: Comparison of different exploration schemes. The background color shows the real altitude of the terrain. All algorithms are run for 525 iterations, or until the first unsafe action is attempted. The saturated color indicates the region that each strategy is able to explore. The baselines get stuck in the crater in the bottom-right corner or fail to explore, while Algorithm 1 manages to safely explore the unknown environment. See the statistics in Fig. 2f.

We compare these baselines over an 120 by 70 meters area at $-30.6°$ latitude and $202.2°$ longitude. We set the accuracy $\epsilon = \sigma_n \beta$. The resulting exploration behaviors can be seen in Fig. 2. The rover starts in the center-top part of the plot, a relatively planar area. In the top-right corner there is a hill that the rover cannot climb, while in the bottom-right corner there is a crater that, once entered, the rover cannot leave. The safe behavior that we expect is to explore the planar area, without moving into the crater or attempting to climb the hill. We run all algorithms for 525 iterations or until the first unsafe action is attempted. It can be seen in Fig. 2e that our method explores the safe area that surrounds the crater, without attempting to move inside. While some state-action pairs closer to the crater are also safe, the GP model would require more data to classify them as safe with the necessary confidence. In contrast, the baselines perform significantly worse. The baseline that does not ensure the ability to return to the safe set (non-ergodic) can be seen in Fig. 2a. It does not explore the area, because it quickly reaches a state without a safe path to the next target sample. Our approach avoids these situations explicitly. The unsafe exploration baseline in Fig. 2b considers ergodicity, but concludes that every state is reachable according to the MDP model. Consequently, it follows a path that crosses the boundary of the crater and eventually evaluates an unsafe action. Overall, it is not enough to consider only ergodicity or only safety, in order to solve the safe exploration problem. The random exploration in Fig. 2c attempts an unsafe action after some exploration. In contrast, Algorithm 1 manages to safely explore a large part of the unknown environment. Running the algorithm without considering expanders leads to the behavior in Fig. 2d, which is safe, but only manages to explore a small subset of the safely reachable area within the same number of iterations in which Algorithm 1 explores over $80\%$ of it. The results are summarized in Table 2f.

# 6 Conclusion

We presented SAFEMDP, an algorithm to safely explore *a priori* unknown environments. We used a Gaussian process to model the safety constraints, which allows the algorithm to reason about the safety of state-action pairs before visiting them. An important aspect of the algorithm is that it considers the transition dynamics of the MDP in order to ensure that there is a safe return route before visiting states. We proved that the algorithm is capable of exploring the full safely reachable region with few measurements, and demonstrated its practicality and performance in experiments.

**Acknowledgement.** This research was partially supported by the Max Planck ETH Center for Learning Systems and SNSF grant 200020_159557.

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
