[Supplementary Material]

# A Preliminary lemmas

**Lemma 3.** $\forall \mathbf{s} \in \mathcal{S}$, $u_{t+1}(\mathbf{s}) \leq u_t(\mathbf{s})$, $l_{t+1}(\mathbf{s}) \geq l_t(\mathbf{s})$, $w_{t+1}(\mathbf{s}) \leq w_t(\mathbf{s})$.

*Proof.* This lemma follows directly from the definitions of $u_t(\mathbf{s})$, $l_t(\mathbf{s})$, $w_t(\mathbf{s})$ and $C_t(\mathbf{s})$. $\square$

**Lemma 4.** $\forall n \geq 1$, $\mathbf{s} \in R_n^{\mathrm{ret}}(\overline{S}, S) \implies \mathbf{s} \in S \cup \overline{S}$.

*Proof.* Proof by induction. Consider $n = 1$, then $\mathbf{s} \in R^{\mathrm{ret}}(\overline{S}, S) \implies \mathbf{s} \in S \cup \overline{S}$ by definition. For the induction step, assume $\mathbf{s} \in R_{n-1}^{\mathrm{ret}}(\overline{S}, S) \implies \mathbf{s} \in S \cup \overline{S}$. Now consider $\mathbf{s} \in R_n^{\mathrm{ret}}(\overline{S}, S)$. We know that

$$R_n^{\mathrm{ret}}(\overline{S}, S) = R^{\mathrm{ret}}(\overline{S}, R_{n-1}^{\mathrm{ret}}(\overline{S}, S)),$$
$$= R_{n-1}^{\mathrm{ret}}(\overline{S}, S) \cup \{\mathbf{s} \in \overline{S} \,|\, \exists a \in \mathcal{A}(\mathbf{s}) \colon f(\mathbf{s}, a) \in R_{n-1}^{\mathrm{ret}}(\overline{S}, S)\}.$$

Therefore, since $\mathbf{s} \in R_{n-1}^{\mathrm{ret}}(\overline{S}, S) \implies \mathbf{s} \in S \cup \overline{S}$ and $\overline{S} \subseteq \overline{S} \cup S$, it follows that $\mathbf{s} \in S \cup \overline{S}$ and the induction step is complete. $\square$

**Lemma 5.** $\forall n \geq 1$, $\mathbf{s} \in R_n^{\mathrm{ret}}(\overline{S}, S) \iff \exists k, 0 \leq k \leq n$ *and* $(a_1, \ldots, a_k)$, *a sequence of $k$ actions, that induces* $(\mathbf{s}_0, \mathbf{s}_1, \ldots, \mathbf{s}_k)$ *starting at* $\mathbf{s}_0 = \mathbf{s}$, *such that* $\mathbf{s}_i \in \overline{S}$, $\forall i = 0, \ldots, k-1$ *and* $\mathbf{s}_k \in S$.

*Proof.* ( $\implies$ ). $\mathbf{s} \in R_n^{\mathrm{ret}}(\overline{S}, S)$ means that either $\mathbf{s} \in R_{n-1}^{\mathrm{ret}}(\overline{S}, S)$ or $\exists a \in \mathcal{A}(\mathbf{s}) \colon f(\mathbf{s}, a) \in R_{n-1}^{\mathrm{ret}}(\overline{S}, S)$. Therefore, we can reach a state in $R_{n-1}^{\mathrm{ret}}(\overline{S}, S)$ taking at most one action. Repeating this procedure $i$ times, the system reaches a state in $R_{n-i}^{\mathrm{ret}}(\overline{S}, S)$ with at most $i$ actions. In particular, if we choose $i = n$, we prove the agent reaches $S$ with at most $n$ actions. Therefore there is a sequence of actions of length $k$, with $0 \leq k \leq n$, inducing a state trajectory such that: $\mathbf{s}_0 = \mathbf{s}$, $\mathbf{s}_i \in R_{n-i}^{\mathrm{ret}}(\overline{S}, S) \subseteq \overline{S} \cup S$ for every $i = 0, \ldots, k-1$ and $\mathbf{s}_k \in S$.
( $\impliedby$ ). Consider $k = 0$. This means that $\mathbf{s} \in S \subseteq R_n^{\mathrm{ret}}(\overline{S}, S)$. In case $k = 1$ we have that $\mathbf{s}_0 \in \overline{S}$ and that $f(\mathbf{s}_0, a_1) \in S$. Therefore $\mathbf{s} \in R^{\mathrm{ret}}(\overline{S}, S) \subseteq R_n^{\mathrm{ret}}(\overline{S}, S)$. For $k \geq 2$ we know $\mathbf{s}_{k-1} \in \overline{S}$ and $f(\mathbf{s}_{k-1}, a_k) \in S \implies \mathbf{s}_{k-1} \in R^{\mathrm{ret}}(\overline{S}, S)$. Similarly $\mathbf{s}_{k-2} \in \overline{S}$ and $f(\mathbf{s}_{k-2}, a_{k-1}) = \mathbf{s}_{k-1} \in R^{\mathrm{ret}}(\overline{S}, S) \implies \mathbf{s}_{k-2} \in R_2^{\mathrm{ret}}(\overline{S}, S)$. For any $0 \leq k \leq n$ we can apply this reasoning $k$ times and prove that $\mathbf{s} \in R_k^{\mathrm{ret}}(\overline{S}, S) \subseteq R_n^{\mathrm{ret}}(\overline{S}, S)$. $\square$

**Lemma 6.** $\forall \overline{S}, S \subseteq \mathcal{S}$, $\forall N \geq |\mathcal{S}|$, $R_N^{\mathrm{ret}}(\overline{S}, S) = R_{N+1}^{\mathrm{ret}}(\overline{S}, S) = \overline{R}^{\mathrm{ret}}(\overline{S}, S)$

*Proof.* This is a direct consequence of Lemma 5. In fact, Lemma 5 states that $\mathbf{s}$ belongs to $R_N^{\mathrm{ret}}(\overline{S}, S)$ if and only if there is a path of length at most $N$ starting from $\mathbf{s}$ contained in $\overline{S}$ that drives the system to a state in $S$. Since we are dealing with a finite MDP, there are $|\mathcal{S}|$ different states. Therefore, if such a path exists it cannot be longer than $|\mathcal{S}|$. $\square$

**Lemma 7.** *Given $S \subseteq R \subseteq \mathcal{S}$ and $\overline{S} \subseteq \overline{R} \subseteq \mathcal{S}$, it holds that $\overline{R}^{\mathrm{ret}}(\overline{S}, S) \subseteq \overline{R}^{\mathrm{ret}}(\overline{R}, R)$.*

*Proof.* Let $\mathbf{s} \in \overline{R}^{\mathrm{ret}}(\overline{S}, S)$. It follows from Lemmas 5 and 6 that there exists a sequence of actions, $(a_1, \ldots, a_k)$, with $0 \leq k \leq |\mathcal{S}|$, that induces a state trajectory, $(\mathbf{s}_0, \mathbf{s}_1, \ldots, \mathbf{s}_k)$, starting at $\mathbf{s}_0 = \mathbf{s}$ with $\mathbf{s}_i \in \overline{S} \subseteq \overline{R}$, $\forall i = 1, \ldots, k-1$ and $s_k \in S \subseteq R$. Using the ( $\impliedby$ ) direction of Lemma 5 and Lemma 6, we conclude that $\mathbf{s} \in \overline{R}^{\mathrm{ret}}(\overline{R}, R)$. $\square$

**Lemma 8.** $S \subseteq R \implies R^{\mathrm{reach}}(S) \subseteq R^{\mathrm{reach}}(R)$.

*Proof.* Consider $\mathbf{s} \in R^{\mathrm{reach}}(S)$. Then either $\mathbf{s} \in S \subseteq R$ or $\exists \hat{s} \in S \subseteq R$, $\hat{a} \in \mathcal{A}(\hat{s}) \colon \mathbf{s} = f(\hat{s}, \hat{a})$, by definition. This implies that $\mathbf{s} \in R^{\mathrm{reach}}(R)$. $\square$

**Lemma 9.** *For any $t \geq 1$, $S_0 \subseteq S_t \subseteq S_{t+1}$ and $\hat{S}_0 \subseteq \hat{S}_t \subseteq \hat{S}_{t+1}$*

*Proof.* Proof by induction. Consider $\mathbf{s} \in S_0$, $S_0 = \hat{S}_0$ by initialization. We known that

$$l_1(\mathbf{s}) - Ld(\mathbf{s}, \mathbf{s}) = l_1(\mathbf{s}) \geq l_0(\mathbf{s}) \geq h,$$

where the last inequality follows from Lemma 3. This implies that $\mathbf{s} \in S_1$ or, equivalently, that $S_0 \subseteq S_1$. Furthermore, we know by initialization that $\mathbf{s} \in R^{\mathrm{reach}}(\hat{S}_0)$. Moreover, we can say that $\mathbf{s} \in \overline{R}^{\mathrm{ret}}(S_1, \hat{S}_0)$, since $S_1 \supseteq S_0 = \hat{S}_0$. We can conclude that $\mathbf{s} \in \hat{S}_1$. For the induction step assume that $S_{t-1} \subseteq S_t$ and $\hat{S}_{t-1} \subseteq \hat{S}_t$. Let $\mathbf{s} \in S_t$. Then,

$$\exists \mathbf{s}' \in \hat{S}_{t-1} \subseteq \hat{S}_t : l_t(\mathbf{s}') - Ld(\mathbf{s}, \mathbf{s}') \geq h.$$

Furthermore, it follows from Lemma 3 that $l_{t+1}(\mathbf{s}') - Ld(\mathbf{s}, \mathbf{s}') \geq l_t(\mathbf{s}') - Ld(\mathbf{s}, \mathbf{s}')$. This implies that $l_{t+1}(\mathbf{s}') - Ld(\mathbf{s}, \mathbf{s}') \geq h$. Thus $\mathbf{s} \in S_{t+1}$. Now consider $\mathbf{s} \in \hat{S}_t$. We known that

$$\mathbf{s} \in R^{\mathrm{reach}}(\hat{S}_{t-1}) \subseteq R^{\mathrm{reach}}(\hat{S}_t) \qquad \text{by Lemma 8}$$

We also know that $\mathbf{s} \in \overline{R}^{\mathrm{ret}}(S_t, \hat{S}_{t-1})$. Since we just proved that $S_t \subseteq S_{t+1}$ and we assumed $\hat{S}_{t-1} \subseteq \hat{S}_t$ for the induction step, Lemma 7 allows us to say that $\mathbf{s} \in \overline{R}^{\mathrm{ret}}(S_{t+1}, \hat{S}_t)$. All together this allows us to complete the induction step by saying $\mathbf{s} \in \hat{S}_{t+1}$. $\square$

**Lemma 10.** $S \subseteq R \implies R_\epsilon^{\mathrm{safe}}(S) \subseteq R_\epsilon^{\mathrm{safe}}(R)$.

*Proof.* Consider $\mathbf{s} \in R_\epsilon^{\mathrm{safe}}(S)$, we can say that:

$$\exists \mathbf{s}' \in S \subseteq R : r(\mathbf{z}') - \epsilon - Ld(\mathbf{z}, \mathbf{z}') \geq h \qquad (9)$$

This means that $\mathbf{s} \in R_\epsilon^{\mathrm{safe}}(R)$ $\square$

**Lemma 11.** *Given two sets $S, R \subseteq \mathcal{S}$ such that $S \subseteq R$, it holds that: $R_\epsilon(S) \subseteq R_\epsilon(R)$.*

*Proof.* We have to prove that:

$$\mathbf{s} \in (R^{\mathrm{reach}}(S) \cap \overline{R}^{\mathrm{ret}}(R_\epsilon^{\mathrm{safe}}(S), S)) \implies \mathbf{s} \in (R^{\mathrm{reach}}(R) \cap \overline{R}^{\mathrm{ret}}(R_\epsilon^{\mathrm{safe}}(R), R)) \qquad (10)$$

Let's start by checking the reachability condition first:

$$\mathbf{s} \in R^{\mathrm{reach}}(S) \implies \mathbf{s} \in R^{\mathrm{reach}}(R). \qquad \text{by Lemma 8}$$

Now let's focus on the recovery condition. We use Lemmas 7 and 10 to say that $\mathbf{s} \in \overline{R}^{\mathrm{ret}}(R_\epsilon^{\mathrm{safe}}(S), S)$ implies that $\mathbf{s} \in \overline{R}^{\mathrm{ret}}(R_\epsilon^{\mathrm{safe}}(R), R)$ and this completes the proof. $\square$

**Lemma 12.** *Given two sets $S, R \subseteq \mathcal{S}$ such that $S \subseteq R$, the following holds: $\overline{R}_\epsilon(S) \subseteq \overline{R}_\epsilon(R)$.*

*Proof.* The result follows by repeatedly applying Lemma 11. $\square$

**Lemma 13.** *Assume that $\|r\|_k^2 \leq B$, and that the noise $\omega_t$ is zero-mean conditioned on the history, as well as uniformly bounded by $\sigma$ for all $t > 0$. If $\beta_t$ is chosen as in (8), then, for all $t > 0$ and all $\mathbf{s} \in \mathcal{S}$, it holds with probability at least $1 - \delta$ that $|r(\mathbf{s}) - \mu_{t-1}(\mathbf{s})| \leq \beta_t^{\frac{1}{2}} \sigma_{t-1}(\mathbf{s})$.*

*Proof.* See Theorem 6 in [21]. $\square$

**Lemma 1.** *Assume that $\|r\|_k^2 \leq B$, and that the noise $\omega_t$ is zero-mean conditioned on the history, as well as uniformly bounded by $\sigma$ for all $t > 0$. If $\beta_t$ is chosen as in (8), then, for all $t > 0$ and all $\mathbf{s} \in \mathcal{S}$, it holds with probability at least $1 - \delta$ that $r(\mathbf{s}) \in C_t(\mathbf{s})$.*

*Proof.* See Corollary 1 in [22]. $\square$

## B Safety

**Lemma 14.** *For all $t \geq 1$ and for all $\mathbf{s} \in \hat{S}_t$, $\exists \mathbf{s}' \in S_0$ such that $\mathbf{s} \in \overline{R}^{\mathrm{ret}}(S_t, \{\mathbf{s}'\})$.*

*Proof.* We use a recursive argument to prove this lemma. Since $\mathbf{s} \in \hat{S}_t$, we know that $\mathbf{s} \in \overline{R}^{\mathrm{ret}}(S_t, \hat{S}_{t-1})$. Because of Lemmas 5 and 6 we know $\exists (a_1, \ldots, a_j)$, with $j \leq |\mathcal{S}|$, inducing $\mathbf{s}_0, \mathbf{s}_1, \ldots, \mathbf{s}_j$ such that $\mathbf{s}_0 = \mathbf{s}$, $\mathbf{s}_i \in S_t$, $\forall i = 1, \ldots, j-1$ and $s_j \in \hat{S}_{t-1}$. Similarly, we can build another sequence of actions that drives the system to some state in $\hat{S}_{t-2}$ passing through $S_{t-1} \subseteq S_t$ starting from $\mathbf{s}_j \in \hat{S}_{t-1}$. By applying repeatedly this procedure we can build a finite sequence of actions that drives the system to a state $\mathbf{s}' \in S_0$ passing through $S_t$ starting from $\mathbf{s}$. Because of Lemmas 5 and 6 this is equivalent to $\mathbf{s} \in \overline{R}^{\mathrm{ret}}(S_t, \{\mathbf{s}'\})$. $\square$

**Lemma 15.** *For all $t \geq 1$ and for all $\mathbf{s} \in \hat{S}_t$, $\exists \mathbf{s}' \in S_0$ such that $\mathbf{s}' \in \overline{R}^{\mathrm{ret}}(S_t, \{\mathbf{s}\})$.*

*Proof.* The proof is analogous to the the one we gave for Lemma 14. The only difference is that here we need to use the reachability property of $\hat{S}_t$ instead of the recovery property of $\hat{S}_t$. $\square$

**Lemma 2.** *Assume that $S_0 \neq \emptyset$ and that for all states, $\mathbf{s}, \mathbf{s}' \in S_0$, $\mathbf{s} \in \overline{R}^{\mathrm{ret}}(S_0, \{\mathbf{s}'\})$. Then, when using Algorithm 1 under the assumptions in Theorem 1, for all $t > 0$ and for all states, $\mathbf{s}, \mathbf{s}' \in \hat{S}_t$, $\mathbf{s} \in \overline{R}^{\mathrm{ret}}(S_t, \{\mathbf{s}'\})$.*

*Proof.* This lemma is a direct consequence of the properties of $S_0$ listed above (that are ensured by the initialization of the algorithm) and of Lemmas 14 and 15 $\square$

**Lemma 16.** *For any $t \geq 0$, the following holds with probability at least $1 - \delta$: $\forall \mathbf{s} \in S_t$, $r(\mathbf{s}) \geq h$.*

*Proof.* Let's prove this result by induction. By initialization we know that $r(\mathbf{s}) \geq h$ for all $\mathbf{s} \in S_0$. For the induction step assume that for all $\mathbf{s} \in S_{t-1}$ holds that $r(\mathbf{s}) \geq h$. For any $\mathbf{s} \in S_t$, by definition, there exists $\mathbf{z} \in \hat{S}_{t-1} \subseteq S_{t-1}$ such that

$$
\begin{aligned}
h &\leq l_t(z) - Ld(\mathbf{s}, \mathbf{z}), \\
&\leq r(\mathbf{z}) - Ld(\mathbf{s}, \mathbf{z}), && \text{by Lemma 1} \\
&\leq r(\mathbf{s}). && \text{by Lipschitz continuity}
\end{aligned}
$$

This relation holds with probability at least $1 - \delta$ because we used Lemma 1 to prove it. $\square$

**Theorem 2.** *For any state $\mathbf{s}$ along any state trajectory induced by Algorithm 1 on a MDP with transition function $f(\mathbf{s}, a)$, we have, with probability at least $1 - \delta$, that $r(\mathbf{s}) \geq h$.*

*Proof.* Let's denote as $(\mathbf{s}_1^t, \mathbf{s}_2^t, \ldots, \mathbf{s}_k^t)$ the state trajectory of the system until the end of iteration $t \geq 0$. We know from Lemma 2 and Algorithm 1 that the $\mathbf{s}_i^t \in S_t$, $\forall i = 1, \ldots, k$. Lemma 16 completes the proof as it allows us to say that $r(\mathbf{s}_i^t) \geq h$, $\forall i = 1, \ldots, k$ with probability at least $1 - \delta$. $\square$

## C Completeness

**Lemma 17.** *For any $t_1 \geq t_0 \geq 1$, if $\hat{S}_{t_1} = \hat{S}_{t_0}$, then, $\forall t$ such that $t_0 \leq t \leq t_1$, it holds that $G_{t+1} \subseteq G_t$*

*Proof.* Since $\hat{S}_t$ is not changing we are always computing the enlargement function over the same points. Therefore we only need to prove that the enlargement function is non increasing. We known from Lemma 3 that $u_t(\mathbf{s})$ is a non increasing function of $t$ for all $\mathbf{s} \in \mathcal{S}$. Furthermore we know that $(\mathcal{S} \setminus S_t) \supseteq (\mathcal{S} \setminus S_{t+1})$ because of Lemma 9. Hence, the enlargement function is non increasing and the proof is complete. $\square$

**Lemma 18.** *For any $t_1 \geq t_0 \geq 1$, if $\hat{S}_{t_1} = \hat{S}_{t_0}$, $C_1 = 8/log(1 + \sigma^{-2})$ and $\mathbf{s}_t = \underset{\mathbf{s} \in G_t}{\operatorname{argmax}}\, w_t(\mathbf{s})$, then, $\forall \bar{t}$ such that $t_0 \leq \bar{t} \leq t_1$, it holds that $w_{\bar{t}}(\mathbf{s}_{\bar{t}}) \leq \sqrt{\frac{C_1 \beta_{\bar{t}} \gamma_{\bar{t}}}{\bar{t} - t_0}}$.*

*Proof.* See Lemma 5 in [22]. $\qquad \square$

**Lemma 19.** *For any $t \geq 1$, if $C_1 = 8/log(1 + \sigma^{-2})$ and $T_t$ is the smallest positive integer such that $\frac{T_t}{\beta_{t+T_t} \gamma_{t+T_t}} \geq \frac{C_1}{\epsilon^2}$ and $S_{t+T_t} = S_t$, then, for any $\mathbf{s} \in G_{t+T_t}$ it holds that $w_{t+T_t}(\mathbf{s}) \leq \epsilon$*

*Proof.* The proof is trivial because $T_t$ was chosen to be the smallest integer for which the right hand side of the inequality proved in Lemma 18 is smaller or equal to $\epsilon$. $\qquad \square$

**Lemma 20.** *For any $t \geq 1$, if $\overline{R}_\epsilon(S_0) \setminus \hat{S}_t \neq \emptyset$, then, $R_\epsilon(\hat{S}_t) \setminus \hat{S}_t \neq \emptyset$.*

*Proof.* For the sake of contradiction assume that $R_\epsilon(\hat{S}_t) \setminus \hat{S}_t = \emptyset$. This implies $R_\epsilon(\hat{S}_t) \subseteq \hat{S}_t$. On the other hand, since $\hat{S}_t$ is included in all the sets whose intersection defines $R_\epsilon(\hat{S}_t)$, we know that, $\hat{S}_t \subseteq R_\epsilon(\hat{S}_t)$. This implies that $\hat{S}_t = R_\epsilon(\hat{S}_t)$.
If we apply repeatedly the one step reachability operator on both sides of the equality we obtain $\overline{R}_\epsilon(\hat{S}_t) = \hat{S}_t$. By Lemmas 9 and 12 we know that

$$S_0 = \hat{S}_0 \subseteq \hat{S}_t \implies \overline{R}_\epsilon(S_0) \subseteq \overline{R}_\epsilon(\hat{S}_t) = \hat{S}_t.$$

This contradicts the assumption that $\overline{R}_\epsilon(S_0) \setminus \hat{S}_t \neq \emptyset$. $\qquad \square$

**Lemma 21.** *For any $t \geq 1$, if $\overline{R}_\epsilon(S_0) \setminus \hat{S}_t \neq \emptyset$, then, with probability at least $1 - \delta$ it holds that $\hat{S}_t \subset \hat{S}_{t+T_t}$.*

*Proof.* By Lemma 20 we know that $\overline{R}_\epsilon(S_0) \setminus \hat{S}_t \neq \emptyset$. This implies that $\exists \mathbf{s} \in R_\epsilon(\hat{S}_t) \setminus \hat{S}_t$. Therefore there exists a $\mathbf{s}' \in \hat{S}_t$ such that:
$$r(\mathbf{s}') - \epsilon - Ld(\mathbf{s}, \mathbf{s}') \geq h \tag{11}$$
For the sake of contradiction assume that $\hat{S}_{t+T_t} = \hat{S}_t$. This means that $\mathbf{s} \in \mathcal{S} \setminus \hat{S}_{t+T_t}$ and $\mathbf{s}' \in \hat{S}_{t+T_t}$. Then we have:

$$
\begin{aligned}
u_{t+T_t}(\mathbf{s}') - Ld(\mathbf{s}, \mathbf{s}') &\geq r(\mathbf{s}') - Ld(\mathbf{s}, \mathbf{s}') && \text{by Lemma 13} \\
&\geq r(\mathbf{s}') - \epsilon - Ld(\mathbf{s}, \mathbf{s}') && (12) \\
&\geq h && \text{by equation 11}
\end{aligned}
$$

Assume, for the sake of contradiction, that $\mathbf{s} \in \mathcal{S} \setminus S_{t+T_t}$. This means that $\mathbf{s}' \in G_{t+T_t}$. We know that for any $t \leq \hat{t} \leq t + T_t$ holds that $\hat{S}_{\hat{t}} = \hat{S}_t$, because $\hat{S}_t = \hat{S}_{t+T_t}$ and $\hat{S}_t \subseteq \hat{S}_{t+1}$ for all $t \geq 1$. Therefore we have $\mathbf{s}' \in \hat{S}_{t+T_t-1}$ such that:

$$
\begin{aligned}
l_{t+T_t}(\mathbf{s}') - Ld(\mathbf{s}, \mathbf{s}') &\geq l_{t+T_t}(\mathbf{s}') - r(\mathbf{s}') + \epsilon + h && \text{by equation 11} \\
&\geq -w_{t+T_t}(\mathbf{s}') + \epsilon + h && \text{by Lemma 13} \\
&\geq h && \text{by Lemma 19}
\end{aligned}
$$

This implies that $\mathbf{s} \in S_{t+T_t}$, which is a contradiction. Thus we can say that $\mathbf{s} \in S_{t+T_t}$.
Now we want to focus on the recovery and reachability properties of $\mathbf{s}$ in order to reach the contradiction that $\mathbf{s} \in \hat{S}_{t+T_t}$. Since $\mathbf{s} \in R_\epsilon(\hat{S}_{t+T_t}) \setminus \hat{S}_{t+T_t}$ we know that:

$$\mathbf{s} \in R^{\text{reach}}(\hat{S}_{t+T_t}) = R^{\text{reach}}(\hat{S}_{t+T_t-1}) \tag{13}$$

We also know that $\mathbf{s} \in R_\epsilon(\hat{S}_{t+T_t}) \setminus \hat{S}_{t+T_t} \implies \mathbf{s} \in \overline{R}^{\text{ret}}(R_\epsilon^{\text{safe}}(\hat{S}_{t+T_t}), \hat{S}_{t+T_t})$. We want to use this fact to prove that $\mathbf{s} \in \overline{R}^{\text{ret}}(S_{t+T_t}, \hat{S}_{t+T_t-1})$. In order to do this, we intend to use the result from Lemma 7. We already know that $\hat{S}_{t+T_t-1} = \hat{S}_{t+T_t}$. Therefore we only need to prove

that $R_\epsilon^{\text{safe}}(\hat{S}_{t+T_t}) \subseteq S_{t+T_t}$. For the sake of contradiction assume this is not true. This means $\exists \mathbf{z} \in R_\epsilon^{\text{safe}}(\hat{S}_{t+T_t}) \setminus S_{t+T_t}$. Therefore there exists a $\mathbf{z}' \in \hat{S}_{t+T_t}$ such that:

$$r(\mathbf{z}') - \epsilon - Ld(\mathbf{z}', \mathbf{z}) \geq h \tag{14}$$

Consequently:

$$
\begin{aligned}
u_{t+T_t}(\mathbf{z}') - Ld(\mathbf{z}', \mathbf{z}) &\geq r(\mathbf{z}') - Ld(\mathbf{z}', \mathbf{z}) && \text{by Lemma 13} \\
&\geq r(\mathbf{z}') - \epsilon - d(\mathbf{z}', \mathbf{z}) && (15) \\
&\geq h && \text{by equation 14}
\end{aligned}
$$

Hence $\mathbf{z}' \in G_{t+T_t}$. Since we proved before that $\hat{S}_{t+T_t} = \hat{S}_{t+T_t-1}$, we can say that $\mathbf{z}' \in \hat{S}_{t+T_t-1}$ and that:

$$
\begin{aligned}
l_{t+T_t}(\mathbf{z}') - Ld(\mathbf{z}', \mathbf{z}) &\geq l_{t+T_t}(\mathbf{z}') - r(\mathbf{z}') + \epsilon + h && \text{by equation 14} \\
&\geq -w_{t+T_t}(\mathbf{z}') + \epsilon + h && \text{by Lemma 13} \\
&\geq h && \text{by Lemma 19}
\end{aligned}
$$

Therefore $\mathbf{z} \in S_{t+T_t}$. This is a contradiction. Thus we can say that $R_\epsilon^{\text{safe}}(\hat{S}_{t+T_t}) \subseteq S_{t+T_t}$. Hence:

$$\mathbf{s} \in R_\epsilon(\hat{S}_{t+T_t}) \setminus \hat{S}_{t+T_t} \implies \mathbf{s} \in \overline{R}^{\text{ret}}(S_{t+T_t}, \hat{S}_{t+T_t-1}) \tag{16}$$

In the end the fact that $\mathbf{s} \in S_{t+T_t}$ and (13) and (16) allow us to conclude that $\mathbf{s} \in \hat{S}_{t+T_t}$. This contradiction proves the theorem. $\qquad\square$

**Lemma 22.** $\forall t \geq 0$, $\hat{S}_t \subseteq \overline{R}_0(S_0)$ with probability at least $1 - \delta$.

*Proof.* Proof by induction. We know that $\hat{S}_0 = S_0 \subseteq \overline{R}_0(S_0)$ by definition. For the induction step assume that for some $t \geq 1$ holds that $\hat{S}_{t-1} \subseteq \overline{R}_0(S_0)$. Our goal is to show that $\mathbf{s} \in \hat{S}_t \implies \mathbf{s} \in \overline{R}_0(S_0)$. In order to this, we will try to show that $\mathbf{s} \in R_0(\hat{S}_{t-1})$. We know that:

$$\mathbf{s} \in \hat{S}_t \implies \mathbf{s} \in R^{\text{reach}}(\hat{S}_{t-1}) \tag{17}$$

Furthermore we can say that:

$$\mathbf{s} \in \hat{S}_t \implies \mathbf{s} \in \overline{R}^{\text{ret}}(S_t, \hat{S}_{t-1}) \tag{18}$$

For any $\mathbf{z} \in S_t$, we know that $\exists \mathbf{z}' \in \hat{S}_{t-1}$ such that:

$$
\begin{aligned}
h &\leq l_t(\mathbf{z}') - Ld(\mathbf{z}, \mathbf{z}'), && (19) \\
&\leq r(\mathbf{z}') - Ld(\mathbf{z}, \mathbf{z}'). && \text{by Lemma 1}
\end{aligned}
$$

This means that $\mathbf{z} \in S_t \implies \mathbf{z} \in R_0^{\text{safe}}(\hat{S}_{t-1})$, or, equivalently, that $S_t \subseteq R_0^{\text{safe}}(\hat{S}_{t-1})$. Hence, Lemma 7 and (18) allow us to say that $\overline{R}^{\text{ret}}(S_t, \hat{S}_{t-1}) \subseteq \overline{R}^{\text{ret}}(R_0^{\text{safe}}(\hat{S}_{t-1}), \hat{S}_{t-1})$. This result, together with (17), leads us to the conclusion that $\mathbf{s} \in R_0(\hat{S}_{t-1})$. We assumed for the induction step that $\hat{S}_{t-1} \subseteq \overline{R}_0(S_0)$. Applying on both sides the set operator $R_0(\cdot)$, we conclude that $R_0(\hat{S}_{t-1}) \subseteq \overline{R}_0(S_0)$. This proves that $\mathbf{s} \in \hat{S}_t \implies \mathbf{s} \in \overline{R}_0(S_0)$ and the induction step is complete. $\qquad\square$

**Lemma 23.** Let $t^*$ be the smallest integer such that $t^* \geq |\overline{R}_0(S_0)|T_{t^*}$, then there exists a $t_0 \leq t^*$ such that, with probability at least $1 - \delta$ holds that $\hat{S}_{t_0+T_{t_0}} = \hat{S}_{t_0}$.

*Proof.* For the sake of contradiction assume that the opposite holds true: $\forall t \leq t^*$, $\hat{S}_t \subset \hat{S}_{t+T_t}$. This implies that $\hat{S}_0 \subset \hat{S}_{T_0}$. Furthermore we know that $T_t$ is increasing in $t$. Therefore $0 \leq t^* \implies T_0 \leq T_{t^*} \implies \hat{S}_{T_0} \subseteq \hat{S}_{T_{t^*}}$. Now if $|\overline{R}_0(S_0)| \geq 1$ we know that:

$$
\begin{aligned}
t^* &\geq T_{t^*} \\
\implies T_{t^*} &\geq T_{T_{t^*}} \\
\implies T_{t^*} + T_{T_{t^*}} &\leq 2T_{t^*} \\
\implies \hat{S}_{T_{t^*}+T_{T_{t^*}}} &\subseteq \hat{S}_{2T_{t^*}}
\end{aligned}
$$

This justifies the following chain of inclusions:

$$\hat{S}_0 \subset \hat{S}_{T_0} \subseteq \hat{S}_{T_{t^*}} \subset \hat{S}_{T_{t^*} + T_{T_{t^*}}} \subseteq \hat{S}_{2T_{t^*}} \subset \dots$$

This means that for any $0 \le k \le |\overline{R}_0(S_0)|$ it holds that $|\hat{S}_{kT_{t^*}}| > k$. In particular, for $k^* = |\overline{R}_0(S_0)|$ we have $|\hat{S}_{k^*T_{t^*}}| > |\overline{R}_0(S_0)|$. This contradicts Lemma 22 (which holds true with probability at least $1 - \delta$). $\qquad\square$

**Lemma 24.** *Let $t^*$ be the smallest integer such that $\frac{t^*}{\beta_{t^*} \gamma_{t^*}} \ge \frac{C_1 |\overline{R}_0(S_0)|}{\epsilon^2}$, then, there is $t_0 \le t^*$ such that $\hat{S}_{t_0 + T_{t_0}} = \hat{S}_{t_0}$ with probability at least $1 - \delta$.*

*Proof.* The proof consists in applying the definition of $T_t$ to the condition of Lemma 23. $\qquad\square$

**Theorem 3.** *Let $t^*$ be the smallest integer such that $\frac{t^*}{\beta_{t^*} \gamma_{t^*}} \ge \frac{C_1 |\overline{R}_0(S_0)|}{\epsilon^2}$, with $C_1 = 8/log(1 + \sigma^{-2})$, then, there is $t_0 \le t^*$ such that $\overline{R}_\epsilon(S_0) \subseteq \hat{S}_{t_0} \subseteq \overline{R}_0(S_0)$ with probability at least $1 - \delta$.*

*Proof.* Due to Lemma 24, we know that $\exists t_0 \le t^*$ such that $\hat{S}_{t_0} = \hat{S}_{t_0 + T_{t_0}}$ with probability at least $1 - \delta$. This implies that $\overline{R}_\epsilon(S_0) \setminus (\hat{S}_t) = \emptyset$ with probability at least $1 - \delta$ because of Lemma 21. Therefore $\overline{R}_\epsilon(S_0) \subseteq \hat{S}_t$. Furthermore we know that $\hat{S}_t \subseteq \overline{R}_0(S_0)$ with probability at least $1 - \delta$ because of Lemma 22 and this completes the proof.

$\qquad\square$

# D  Main result

**Theorem 1.** *Assume that $r(\cdot)$ is L-Lipschitz continuous and that the assumptions of Lemma 1 hold. Also, assume that $S_0 \ne \emptyset$, $r(\mathbf{s}) \ge h$ for all $\mathbf{s} \in S_0$, and that for any two states, $\mathbf{s}, \mathbf{s}' \in S_0$, $\mathbf{s}' \in \overline{R}^{\mathrm{ret}}(S_0, \{\mathbf{s}\})$. Choose $\beta_t$ as in (8). Then, with probability at least $1 - \delta$, we have $r(\mathbf{s}) \ge h$ for any $\mathbf{s}$ along any state trajectory induced by Algorithm 1 on an MDP with transition function $f(\mathbf{s}, a)$. Moreover, let $t^*$ be the smallest integer such that $\frac{t^*}{\beta_{t^*} \gamma_{t^*}} \ge \frac{C |\overline{R}_0(S_0)|}{\epsilon^2}$, with $C = 8/\log(1 + \sigma^{-2})$. Then there exists a $t_0 \le t^*$ such that, with probability at least $1 - \delta$, $\overline{R}_\epsilon(S_0) \subseteq \hat{S}_{t_0} \subseteq \overline{R}_0(S_0)$.*

*Proof.* This is a direct consequence of Theorem 2 and Theorem 3. $\qquad\square$