[Reviews · NeurIPS 2016]

Reviewer 1

Summary

The paper presents an algorithm for exploring deterministic finite-state finite-action MDP while ensuring that the agent never visits unsafe states. The core idea is based on assuming that the reward (or safety) function is Lipschitz with constant L, and can be represented by a GP. The posterior and variance of the GP are used to define upper and lower bounds on the safety of any state. The set of safe states is iteratively expanded by adding new safe states. A new safe state is a state that can be reached from the previous safe states and can return to them, and is guaranteed (with high probability) to have a reward above a given threshold. The GP is updated by inquiring about the value of the state that has the highest difference between its upper and lower bound. The paper also provide a proof that the algorithm explores the maximum reachable safe set without visiting unsafe states (with a high probability).

Qualitative Assessment

The paper is well-written and clear. The proposed idea is interesting. I have the following comments/questions: 1) Does the Liptschiz assumption hold here with a probability or is it assumed to always hold? 2) Figure 1: should it be \bar{s}_2 instead of s_2 in the caption? The use of bar for non-sets is confusing. 3) About the safety requirements: why do you need to be able to return to safe states if you are always in a safe state? I do not see the need for the last intersection in Equation 4. 4) When you repeatedly apply Equation 4, the number of states that satisfy the safety constraint shrinks because you use Liptschiz in the worst scenario sense. Eventually, you will cover all the safe states, but I have the impression that this could be a slow process. I don't know what would be the alternative, perhaps local Liptschiz constants? 5) How do we know the Lipschitz constant L in a real-world problem? It seems to me that this algorithm will not be of much use in practice because we rarely know L beforehand. If you just take the highest possible value for L, then you will suffer from slow exploration. 6) The Lipschitz assumption does not hold in many applications. Often, the safety function has an infinite L, examples include hitting obstacles, joint limit singularity in robotic manipulation, etc. 7) Is the final path that the robot follows optimal, besides being safe? In the proposed algorithm, the robot may end up oscillating between two distant extremes in the state space. 8) Is the inquiry y_t=r(s_t) + noise obtained by actually visiting state s_t or is it calculated just from a model? I think the answer to this question is tied to the previous question. The basic problem here is: is the safety function physically measured, and hence known only after visiting the states unless the function is Lipschitz? or is it obtained from simulation in which case there is no problem in inquiring as many samples as possible? Let's say I have a robotic arm and I do not know it's joint limits, how can I use your algorithm to perform some task?

Confidence in this Review

2-Confident (read it all; understood it all reasonably well)


Reviewer 2

Summary

This paper addresses the topic of safe exploration. This is an important topic and the community is still figuring out the right way to think about this and talk about it, thus different papers wil make different assumptions and there may be some debate about the reasonableness of these assumptions. This paper focuses on what is basically learning a reward function for deterministic MDP using a GP.

Qualitative Assessment

Most of my concerns have been mentioned above. This isn't bad work and I'm not opposed including it in the conference. I'm just struck with the impression that it could possibly make weaker assumptions or use something other than GPs that makes more direct use of the strong assumptions made in the paper.

Confidence in this Review

2-Confident (read it all; understood it all reasonably well)


Reviewer 3

Summary

This paper provides a GP based model to model the safety constrains in a unknown environment, where a safe routine exploration is analyzed in detail.

Qualitative Assessment

1. It would be helpful if the authors provide a definition of the 'safety'. Since this paper focuses on the safe exploration, it is important to know this definition. 2. Could you explain why the safe states are relevant to the rewards? 3. What is \bar{S} in (3)? 4. Usually, a RL algorithm will try to find a policy such that a long-term return is maximized. While this paper studies the safe exploration, it will be interesting to discuss the connection of this exploration with a RL algorithm.

Confidence in this Review

2-Confident (read it all; understood it all reasonably well)


Reviewer 4

Summary

The authors present a method of safely exploring environments using finite Markov decision processes. The proposed method uses a Gaussian process prior on the safety constraints. They provide theoretical analysis on the safety and complete exploration of their algorithm as well as experimental analysis of a simulated Mars rover scenario, demonstrating the superiority of their method over other approaches.

Qualitative Assessment

The paper is clear and very well presented. The figures are informative and well made, and each have significant purpose in the paper. My first main criticism of this paper is that, in the Contribution section starting on line 60, there is mention of the "vastly reduced computational cost" as well as "making computations more tractable," while these points are not emphasized later in the paper or analyzed in the experiments section. The second main criticism is that there is no comparison with [14] in the experiments (or any other reasonable competitor to the proposed method.) The experiments seem to only point out the importance of various aspects of the proposed method.

Confidence in this Review

2-Confident (read it all; understood it all reasonably well)


Reviewer 5

Summary

The contribution of this paper lies in extending the work of safe exploration in Bayesian optimization in [22] to include reachability and returnability constraints within the state space. The theoretical results appear to be sound. From the paper, it is not clear to me though what the non-trivial technical challenges are in extending the proofs and derivations in [22] to account for the additional reachability and returnability constraints. It is also not clear to me how the trade-off of performance in exploring the maximum reachable safe set for computational efficiency for the proposed algorithm compares to that of [14] since there is no empirical comparison with [14] even though a similar setup to [14] is used. I’m also concerned about how sensitive the safe exploration performance of the proposed algorithm is to poorly estimated GP hyperparameter values which are not available a priori. More detailed comments are available below.

Qualitative Assessment

NOVELTY Lines 57-59: The authors say that “[22] considered a bandit setting, where at each iteration any arm can be played. In contrast, in this paper we consider an MDP, which introduces restrictions in terms of reachability that have not been considered in Bayesian optimization before” and “The main contribution consists of extending the work on safe Bayesian optimization in [22] from the bandit setting to MDPs.” What is the implication of such additional reachability and returnability constraints on the difference in the resulting technical proofs and derivations? Does it invalidate the proof technique in [22] or did the authors use a similar proof technique in [22] with additional non-trivial technical challenges to be tackled? PRACTICAL SIGNIFICANCE Lines 281-2: How do you obtain the values of the lengthscales and noise standard deviation hyperparameters, considering that the height distribution is unknown? If you set them manually, how sensitive are your results in terms of safety to different hyperparameter values? Can you justify why it is practical to do this for safe exploration? The authors have used the same experimental setup as that in [14]. Though the authors have indicated that [14] is much more computational demanding than their proposed approach, it is not clear whether how the trade-off of performance in exploring the maximum reachable safe set for computational efficiency for the proposed algorithm compares to that of [14]. Line 78: Can the assumption of a deterministic transition model be relaxed in the proposed work? If so, what are the implications? Does [14] impose a similar assumption? In MDP planning, besides the exploration aspect, there is also the exploitation aspect that is not accounted for nor discussed in this paper. Must the safe exploration in Algorithm 1 be performed separately first before exploiting the MDP for optimal planning? Or can the exploration be performed simultaneously with exploitation in planning like those in the following references (albeit without safety constraints/considerations): C. K. Ling, K. H. Low, and P. Jaillet (2016). Gaussian Process Planning with Lipschitz Continuous Reward Functions: Towards Unifying Bayesian Optimization, Active Learning, and Beyond. In Proc. AAAI. Marchant, R.; Ramos, F.; and Sanner, S. (2014). Sequential Bayesian optimisation for spatial-temporal monitoring. In Proc. UAI. CLARITY OF PRESENTATION Equation 1: The authors can give an intuitive explanation of how epsilon is related to noise omega_t and/or the GP predictive uncertainty of r due to limited measurements. To avoid confusion, I recommend naming S_t in line 159 and equation 5 differently from that in line 152 because they are defined differently. In Theorem 1, what value is delta set to? Minor issues: Fig. 1: The notations appear to be inconsistent with that in equation 3. Missing bracket in first line of for loop in Algorithm 1. line 192: Should u_t(s) be u_t(s’) instead? line 194: has has?

Confidence in this Review

2-Confident (read it all; understood it all reasonably well)